# How to Use Lean Thinking for the Optimization of Clinical Pathways: A Systematic Review and a Proposed Framework to Analyze Pathways on a System Level

**DOI:** 10.3390/healthcare11182488

**Published:** 2023-09-07

**Authors:** Joanna R. G. Vijverberg, Marc B. V. Rouppe van der Voort, Paul B. van der Nat, Machteld J. Mosselman, Sander Rigter, Douwe H. Biesma, Frits van Merode

**Affiliations:** 1Care and Public Health Research Institute, Maastricht University, 6200 MD Maastricht, The Netherlands; 2Maastricht University Medical Centre+, 6229 HX Maastricht, The Netherlands; 3Department of Value Improvement, St. Antonius Hospital, 3435 CM Nieuwegein, The Netherlands; 4Department of Juiste Zorg, Juiste Plaats, Juiste Kosten, St. Antonius Hospital, 3435 CM Nieuwegein, The Netherlands; 5Radboud Institute for Health Sciences, Scientific Center for Quality of Healthcare (IQ Healthcare), Radboud University Medical Center, 6525 EP Nijmegen, The Netherlands; 6Department of Anesthesiology and Intensive Care, St. Antonius Hospital, 3435 CM Nieuwegein, The Netherlands; 7Department of Internal Medicine, Leiden University Medical Center, 2333 ZA Leiden, The Netherlands; 8Department of Internal Medicine, University Medical Center Utrecht, 3584 CX Utrecht, The Netherlands

**Keywords:** lean thinking, clinical pathway, quantification, improvement potential

## Abstract

Lean Thinking and clinical pathways are commonly used concepts to improve healthcare. However, little is known on how to use Lean Thinking for the optimization of pathways or the quantification of both concepts. This study aims to create a framework to analyze pathways with Lean Thinking on a system level, by quantifying the seven wastes, flow and pull. A systematic literature review was performed. Inclusion criteria were the focus of the article on a well-defined group of patients and studied a pathway optimization with Lean Thinking. Data were extracted on measured outcomes, type of intervention and type of researched pathway. Thirty-six articles were included. No articles described the implementation of the Lean Thinking philosophy or studied the development of their people and partners (“4 P” model). Most articles used process optimization tools or problem-solving tools. The majority of the studies focused on process measures. The measures found in the review were used as input for our suggested framework to identify and quantify wastes, flow, and pull in a clinical pathway. The proposed framework can be used to create an overview of the improvement potential of a pathway or to analyze the level of improvement after an enhancement is introduced to a pathway. Further research is needed to study the use of the suggested quantifications.

## 1. Introduction

The pressure on healthcare is increasing due to a growing demand for care and a labor shortage that will further increase in the coming years. This leads to an urgency in healthcare for improvement approaches that free up resources. Approaches that focus on value, like Value-Based Healthcare, Lean Thinking and Patient-Centered Care, are increasingly popular [1,2]. The question is whether these approaches have the intended effect.

### 1.1. Theoretical Background

Lean Thinking is one of the approaches widely used. In addition to Lean manufacturing, the term Lean Thinking was introduced by Womack and Jones to describe the philosophy, tools, and principles [3].

The core principle of Lean Thinking is adding value for a customer. All care provided to patients can be seen as a process, a logical sequence of activities delivered by professionals in a certain way with certain materials with certain equipment and in certain places [4]. In a process, there are value-adding steps and non-value-adding steps. These non-value adding steps or ‘wastes’ can be categorized: overproduction, waiting, conveyance, overprocessing, inventory, motion, and defects. The definitions of these seven wastes according to Liker are listed in Table 1 [5].

The Lean Thinking philosophy has a holistic view, thus focuses on a system level. A methodology to implement system changes is the Shingo Model. This model can be used to transform to a culture of operational excellence [6]. The aim is to improve the hospital and not one department. A hospital which is not yet mature in operational excellence should not start with changing the entire hospital, because this is too large. It could start with local optimization, without compromising other departments. A hospital can progress in the level of optimization by cooperating with an increasing number of departments.

### 1.2. Research on Lean Thinking in Healthcare

In the last decade, multiple systematic reviews were conducted on the implementation of Lean Thinking in healthcare. They showed a lack of empirical evidence [2,7]. The research field is missing a framework that should clarify what is being investigated in accordance with Lean Thinking [7]. This will contribute to the quality of further empirical research.

Three reviews concluded that the current literature consists of studies with a narrow scope and superficial implementation [8,9,10]. Mazzocato et al. formulated it as a “narrow technical application with limited organizational reach [8]”. Another review added that tools that “require a higher degree of knowledge and maturity of healthcare institutions are infrequently used [10]”.

### 1.3. Importance of a Holistic View

Research is mainly focused on single departments or wards [2]. Besides the fact that this is not the vision of Lean Thinking, it is also not beneficial for the entire organization. Ludwig et al. investigated the link between departmental efficiency and hospital efficiency and found that the interests of the entire hospital often differ from the interests of the department [11]. This may lead to local optimization and can lead to an adverse effect on the efficiency of the hospital. The analyses also suggested that there was a negative association between hospitals that have efficient departments and the total efficiency at the hospital level. Since a department is not the correct subsystem, another subsystem is needed.

### 1.4. Clinical Pathways

One of such subsystems used in healthcare management is a clinical pathway. De Bleser et al. defined it as “a method for the patient-care management of a well-defined group of patients during a well-defined period of time [12]”. One of the aims of a clinical pathway is to increase the efficiency in the use of resources [12]. A clinical pathway does not focus on one department but follows patients across several departments. Lean Thinking can be used to study the process steps in a clinical pathway. Identifying and removing wastes will lead to a system with a continuous flow process. Another important ingredient of Lean Thinking is to initiate pull in the system.

### 1.5. Quantifying Lean Thinking

To fill the current research gaps (lack of empirical evidence, narrow scope, superficial implementation), a framework to analyze Lean Thinking on a system level is needed. To analyze and understand a system, it is necessary to map it quantitatively, which can show an organization where objectives are not being achieved. The framework can form a basis for conducting empirical research. This lack of knowledge leads to the objectives of this paper.

### 1.6. Objectives

The aim of this study is to create a framework to analyze clinical pathways with Lean Thinking on a system level, by using quantified measurements of the seven wastes, flow and pull.

The following questions were answered:What is known in the current literature on Lean Thinking when used in clinical pathways?What definitions can be used to systemically quantify value (waste, flow, push/pull) in a clinical pathway?

## 2. Methods

### 2.1. Eligibility Criteria

Studies were selected based on the following criteria:Study context: Focused on a clinical pathway in healthcare.Clinical pathway definitions: The pathway was focused on a well-defined group of patients. The paper had to study a pathway optimization with Lean Thinking.Lean Thinking application: The authors explicitly mentioned using Lean for optimization in the methodology.Study types: All study designs were considered, but only peer-reviewed articles were included.Report language: Published in English.Publication years: All publication years were considered eligible for this review.

### 2.2. Information Sources

Literature search strategies were developed for the electronic databases PubMed, EMBASE and Web of Science. These three databases were chosen to provide a good coverage on medical research (PubMed) and healthcare innovation (EMBASE and Web of Science). The search was conducted in November 2022. The reference lists of included articles were scanned to ensure that all relevant studies were included.

### 2.3. Search Strategy

The search strategy for PubMed is listed below. A similar strategy was adapted for each database. The strategy consists of two parts: a strategy to search for articles studying (1) Lean Thinking and (2) clinical pathways. The first part consists of terms linked to Lean Thinking. We use the term Toyota in addition to the terms for Lean because Toyota and Toyota Production System (TPS) often refer to the philosophy developed by the manufacturer Toyota.


*(“Lean Six Sigma”[Title/Abstract] OR “lean thinking”[Title/Abstract] OR “lean”[Title] OR “lean health*”[Title/Abstract] OR “Toyota”[Title/Abstract] OR “lean principles”[Title/Abstract] OR “lean process”[Title/Abstract] OR “lean methodology”[Title/Abstract] OR “lean techniques”[Title/Abstract] OR “lean management”[Title/Abstract])*



*AND*



*(“Clinical pathway”[Title/Abstract] OR “Critical pathway”[Title/Abstract] OR “Care pathway*”[Title/Abstract] OR “Care process”[Title/Abstract] OR “pathway optimization”[Title/Abstract] OR “critical pathways”[MeSH Terms])*


### 2.4. Selection Process

All search results were downloaded to EndNote reference manager. Duplicates were removed before screening. Two reviewers (JRGV, MM) independently screened titles and abstract and selected articles they deem eligible for inclusion. As a second step, the same reviewers assessed the full text for eligibility. After this process, discrepancies between reviewers were resolved in a consensus meeting. The selection process was conducted in the web app Rayyan, which is an online tool for screening articles [13].

### 2.5. Data Collection Process

Data were extracted and collected with Microsoft Excel v16.76 by two reviewers (JRGV, MM). One reviewer (JRGV) checked the collected data for discrepancies.

### 2.6. Data Items

Data were obtained on the main characteristics of the article (year of publication, study design, population size, researched patient group), definitions used for Lean Thinking and clinical pathways, measured outcomes, type of intervention and level of researched pathway (hospital, disease, discipline, intervention).

### 2.7. Analysis and Presentation of Results

The PRISMA flow chart was used to depict the review decision process. The extracted data were summarized quantitatively. The categorical data was expressed as frequencies (%). The analysis was performed with the use of Microsoft Excel v16.76.

### 2.8. Development Proposed Framework

The measurements found in the systematic review were linked to Lean Thinking and contributed to the suggested framework with measurements for the seven wastes, flow and pull. These measurements can be applied to quantify clinical pathways. A clinical pathway has three levels: (1) the entire value stream (all the process steps or activities a patient undergoes for a specific disease from referral to end of treatment [14]); (2) different process trajectories like diagnostic or treatment process (all the process steps a patient undergoes in a specific process); and (3) a process step (a specific step in the value stream performed by one healthcare professional). Waste can be measured on each level of the pathway. This suggested framework focuses on waste on the level of the value stream to keep a system perspective.

To illustrate the suggested measurements, the colorectal cancer clinical pathway was used. This pathway consists of the patient group with (suspected) colorectal cancer entering the hospital after a referral from the General Practitioner, another specialist, or the population screening.

## 3. Results

### 3.1. Selection

The search yielded 171 records, of which 106 remained after duplicates were removed (Figure 1). The screening of title and abstract resulted in 34 excluded articles. Seventy-two articles were assessed for eligibility, of which 20 articles were not on a specific patient group, 15 articles did not use Lean Thinking in their method and one article did not take place in a care setting. This resulted in the inclusion of 36 articles in total. The full data set with included articles and collected outcomes can be found in Appendix A.

### 3.2. Review Findings

#### 3.2.1. General

The included articles were published between 2010 and 2022 with a range of one to six articles per year (Figure 2). In 2019, most articles were published. All articles were original articles. Patients from the medical specialty orthopedic surgery were most often studied (n = 14, 39%). Cardiology and ophthalmology were studied three times (8%). The other specialties, shown in Table 2, occurred one or two times.

#### 3.2.2. Lean Thinking

In 42 percent of the articles (n = 15) Lean Thinking was used to optimize processes and in 58 percent of the articles (n = 21) a combination of Lean and Six Sigma (LSS) was used. The Lean methods used in the articles were mostly value stream mapping (n = 19 of 63, 30%) and process mapping (n = 12 of 63, 19%). The other methods used are listed in Table 3.

In total, four articles (n = 4 of 36, 11%) used the plan-do-check-act methodology for their process design. Sixteen of the articles that used LSS (n = 16 of 21, 76%) used the steps of the DMAIC-methodology (define, measure, analyze, improve and control) for the process design. The other articles did not use an improvement methodology from Lean Thinking or Six Sigma.

The methods used in the articles were classified in the “4 P” model shown in Table 3. No articles were included that implemented Lean methods in the “philosophy” and “people and partners” category. The majority (n = 23, 64%) of the articles applied methods from the “process” category. Thirty-three percent (n = 12) of the articles applied methods from both the “process” and “problem solving” categories. One article (3%) only applied “problem solving” methods.

#### 3.2.3. Clinical Pathways

The researched patient groups were mostly selected around a disease (n = 24, 67%), a quarter of the patient groups were treatment-based (n = 9, 25%). The rest was once discipline-based, once unit-based and once hospital-based.

The articles focused mostly on a multidisciplinary specialist team (n = 26, 72%).

#### 3.2.4. Measures

The found measures according to the ‘Donabedian’ model [17] were: process measures (n = 65, 66%), outcome measures (n = 29, 30%) and structure measures (n = 0, 0%). Four articles used a cost measure (n = 4, 4%). In two articles, no measures were collected. Multiple measures could be evaluated in one article.

The top three most used process measures were length of stay in the hospital (n = 16, 44%), time between process steps (n = 13, 36%) and duration of certain process steps (n = 10, 28%). The other process measures are listed in Table 4. In one article, the waste in a process was counted. They did not specify waste as a quantitative, but only described what types of waste occurred. The two most measured outcomes were patient satisfaction (n = 5, 14%) and number of readmissions (n = 5, 14%). The other outcome measures are also listed in Table 4.

### 3.3. Linking Found Measures with Lean Thinking

The articles found in our search provided process and outcome measures that contributed to the suggested quantifications of Lean Thinking in a clinical pathway. In Table 5, the found measures were linked to Lean Thinking concepts (seven wastes, flow and pull). For overproduction, transport, movement and pull no links were found in the current literature.

### 3.4. The Framework for Measuring Lean Thinking in a Clinical Pathway

Quantifications for measuring Lean Thinking in a clinical pathway are selected from Table 5 and supplemented with our suggestions to create a framework that consists of nine elements: the seven wastes, flow and pull. For each of these elements the framework contains a quantification. In the following section, the elements are described. In Table 6, the suggested quantifications (variable or formula) are shown in an overview. To illustrate the variables we suggest, Figure 3, Figure 4 and Figure 5 show an example of a clinical pathway. Figure 3 depicts a dummy of a pathway inside the hospital system, Figure 4 zooms in on the clinical pathway to demonstrate which variables can be measured on the level of a step and Figure 5 shows a legend to the figures.

#### 3.4.1. Waste

*Overproduction* is producing items for which there are no orders, resulting in stock build up between processes and producing more than is required [5,18].

Overproduction and pull are closely related. Overproduction occurs when something is produced before someone downstream in the pathway is ready for it or needs it. Overproduction is a symptom of the absence of pull. Pull and push can be explained with a closed and open loop system [19]. A closed loop system only admits new patients if the system has downstream capacity, like in the Kanban system. If there is capacity in the clinical pathway downstream, a signal will trigger the admission of new patients upstream. There is direct feedback between the steps. In a hospital, there is usually no or no timely feedback between downstream and upstream steps between departments used by a clinical pathway. When this feedback is missing patients are admitted regardless of whether there is capacity downstream in the system or patients are not admitted when capacity is actually available. The hospital is open loop and works like a push system. In case of overproduction, patients are allowed into the system while there is no capacity for them downstream. This causes local pile-up.

To measure overproduction in a clinical pathway, we suggest using the number of patients waiting in the system between two steps. In the case of colorectal cancer, an example of overproduction is patients waiting for an operation. The patients waiting for a first appointment after their referral are not waiting in the system and should not be counted as overproduction.

2.In the literature found in this review, waiting time between process steps was often measured and could be an indicator of the waste *waiting*. We suggest quantifying waiting as the time a patient is waiting between two steps in the system. An example is the time between an appointment at the outpatient clinic for an intake with a gastroenterologist and the next step, a diagnostic colonoscopy. Nota bene, sometimes there is useful waiting in a clinical pathway, for instance when a patient needs to revisit the outpatient clinic after six months for follow-up or when a patient needs two days of colon preparation before a colonoscopy.3.*Conveyance* (transport) was not linked to any of the measures found in the literature review. The movement of a patient or materials necessary in the process can often be improved mainly at process step level and is not easily reflected in a clinical pathway. A solution is to indicate the location of each step in the pathway. By measuring the physical distance between the different locations where a patient moves to on a single day, the conveyance can be expressed. In addition, if an outpatient appointment takes place in the hospital instead of digitally the patient experiences extra transport. This should be considered when calculating conveyance.4.With *overprocessing*, there are steps in the process that are unneeded or not the right step or the processing takes unnecessary precision [5,18]. In the literature review, three measures were linked to overprocessing: (1) the duration of a process step; (2) the number of certain process steps in a trajectory; (3) pathway or guideline compliance and protocol utilization. The first is on the level of a process step, the second on the level of a process trajectory and the third is on a value stream map level. The last one can be used in a clinical pathway to observe an undesirable deviation from the pathway. Like when a patient is given an appointment after the colonoscopy for the results and is then referred to another specialist for treatment. These two steps can be combined into one step by giving the patient the results of the colonoscopy by the treating specialist. In this way, the patient does not need to visit the hospital twice and receives information about the possible next treatment steps. We suggest quantifying overprocessing as the number of extra, undesirable steps for a patient that took place compared to the desired pathway. The healthcare professionals who work within the clinical pathway should jointly determine what the desired pathway is.5.Excess *inventory* is idle stock. In healthcare, patients that are waiting for appointments or treatment could be considered as inventory. In this review, the number of patients in certain process steps/trajectory was linked to inventory. To map the inventory in the pathway, the number of patients in a certain trajectory, equivalent to the work in process (WIP), can be measured. For example, all patients waiting for surgery are WIP in this certain trajectory.6.Unnecessary *movement* is motion that does not create added value or searching for information [18]. This is mostly noticeable locally for the staff, but less over the entire clinical pathway. A nurse will need different materials, such as a scale, to measure the weight of the patient. This may not be present at the location where the patient is, and the nurse will get the scale somewhere else. This causes unnecessary movement and makes the step take longer. The extra movement will not cause waste on value stream level but may cause waste at process step level because the step possibly takes longer. Unnecessary movement on process step level could be measured as the number of physical steps taken by a staff member during a process step. We propose to exclude movement when quantifying the clinical pathway because measuring movement does not add value on value stream map level.7.*Defects* are the production of defective parts, and this needs correction [5]. In the literature, we found ten measures to calculate defects. To measure defects on the level of the value stream map, patient satisfaction of the entire pathway, complications, mortality, and functional recovery could be used. These are important outcome measures in creating value for the patient.

#### 3.4.2. Flow and Pull

*Flow* is a continuous process in which there is little waste and a stable output. We linked flow to the found measure: duration of the whole trajectory. To specify flow further, we propose to include waiting time in the calculation. This results in the ratio of total waiting time to the cycle time in the system to quantify flow. Waiting time is defined in the section above. The cycle time is defined as the total duration between the time a patient enters the clinical pathway and the time a patient leaves the clinical pathway. The access time (time between referral and first appointment) is not part of the cycle time.No found measures were linked to *pull*. Munavalli links open and closed loop systems to push and pull systems [20], explained under 1. *overproduction*. In a push system, there is no limit to the work in process (WIP) and no feedback to determine the input, thus an open loop system. The opposite applies for pull systems: the system determines if new input can be admitted and it uses the system status to determine this, thus a closed loop system. The WIP can be used to create a feedback system where new input (a patient) in the clinical pathway is admitted when another patient leaves the system, like CONWIP (constant work in process) [21]. To measure pull in a clinical pathway, the WIP of the entire system should be calculated: the total amount of patients in the system at a given time.

## 4. Discussion

### 4.1. Main Findings

This review showed that the current literature on identifying and quantifying wastes, flow, and pull in a clinical pathway is not extensive. The search found 36 articles using Lean methods to improve clinical pathways. No articles described the implementation of the Lean Thinking philosophy or studied the development of their people and partners (“4 P” model). Most articles used process optimization tools from Lean Thinking, like value stream mapping and waste reduction, or problem-solving tools, like the cause-and-effect diagram and the 5 whys. The use of these tools is embedded at dimension two of the Shingo model (process improvement). Process measures, like length of stay and time between process steps, were mostly studied. We suggested quantifications to identify and quantify wastes, flow, and pull in a clinical pathway and used the process and the few outcome measures that we found in the review as input. The found literature did not contain of quantifications for overproduction, transport, movement, and pull. We made suggestions for these four elements, based on expert opinions and literature outside the healthcare scope. This resulted in a framework that can be used to create an overview of the improvement potential of a clinical pathway or to analyze the level of improvement after an enhancement is introduced to the pathway. Also, identifying the types of waste enables improvement initiatives to select the most relevant Lean principles and tools to potentially reduce more waste and have more positive impact on outcomes.

### 4.2. System Level

Other literature reviews confirmed that the focus is often on a few tools of improvement approaches rather than on the whole philosophy [1,2,8,9,10]. The lack of studies that implement Lean Thinking in a clinical pathway or on a higher organization level than departmental is confirmed by Van der Ham et al. [22]. They reviewed logistical parameters used in hospitals and found that few studies do cross-departmental research. Most articles in the review used logistic parameters that were focused on departments [22]. It is worrying that few studies are done at the system level, because optimizing locally can be at the cost of the system performance [11].

The introduction stated that the Shingo model can be used to implement Lean Thinking systemically. The results of our study showed that Lean Thinking is implemented on dimension two of the Shingo model (process improvement). Unfortunately, dimensions three and four (resp. enterprise alignment and results) have not been explored much. Dimension three is important within healthcare. By viewing the system horizontally (clinical pathways) instead of vertically (departments), more alignment can take place. With a horizontal view, the communality and modularity (part of a system that is shared with different patient groups) of clinical pathways can be examined. This indicates which process steps of different clinical pathways correspond and can be shared and standardized into a trajectory for different patient groups.

### 4.3. Clinical Pathways in Integrated Practice Units

Regardless of what concept a hospital uses, clinical pathways or Integrated Practice Units (IPUs), the suggested framework can be used. A clinical pathway and an IPU are separate concepts in healthcare but could be used simultaneously and benefit from each other. An IPU, which is an important component of value-based healthcare, is a multidisciplinary team working together on a medical condition or set of related conditions. It is stated, by Porter and Lee, that the unit has a “single administrative and scheduling structure” and that the clinicians who work in the unit “devote a significant portion of their time to the medical condition” [23].

Both clinical pathways and IPUs are disease-oriented, multidisciplinary, and horizontally organized. However, clinical pathways focus on standardizing care processes for a specific condition and an IPU focuses on creating an organizational structure to give the best care for a specific group of patients. One or more clinical pathways can be situated within the organizational structure of an IPU.

Integrating clinical pathways within IPUs provides two advantages for analyzing and improving clinical pathways with Lean Thinking. First, an IPU provides a system view because it is cross-departmental. Second, an IPU is suitable for creating a closed loop system.

In a traditional (vertically organized) hospital, a patient with suspected colorectal cancer is waiting for an appointment at a gastroenterology department in line with other patients with various medical conditions, and thus other clinical pathways. The waiting list of this department is completed by a first-come, first-serve principle, and without information about the capacity of other departments, where patients need care later in the clinical pathway. This is an open loop system, without feedback about capacity between departments. This makes it difficult to achieve pull, however it is not impossible.

An IPU is organized horizontally and this theoretically provides an advantage to create a closed loop system, where the system receive feedback from downstream about available capacity for new patients. In addition, IPUs have waiting lists for patients with similar medical conditions, instead of patients in need of an appointment at a similar department. An IPU provides the organizational structure to achieve pull, a constant work in process, by becoming a closed loop system.

### 4.4. Focus on Orthopedic Surgery

Almost 40% of the included articles focused on orthopedic surgery. A possible explanation is the short care pathways for orthopedic diseases. The pathways are more straightforward and often only contain one specialty. In contrast to, for example, colorectal cancer, where a patient is often treated by doctors from multiple specialties and there is a wider variety in treatment options. This focus was also seen in a scoping review on value-based healthcare [1], but other Lean Thinking reviews included studies mostly focused on the emergency department [2,8,24].

### 4.5. Strengths and Limitations

There are a few limitations to this study. First, no literature outside the healthcare sector was reviewed. Perhaps this could have served as inspiration for the development of the suggested quantifications. Other frameworks could have been designed for or used in comparable service organizations. Measurements used in these organizations could provide out-of-the-box solutions or the new insights for healthcare.

Second, there is a widespread application of Lean Thinking in healthcare shown by a study in the United States were 69.3% of the 1222 hospitals responding to a survey reported use of Lean Thinking in their hospital [25]. But in this study only one to six articles per year were found. A possible explanation for this limited number of included articles, may be that practical experiences with quantifying Lean Thinking have not resulted in the publication of scientific articles. The inclusion of grey literature could have enriched the results of the search. The results showed a drop of included articles after 2019, it is possible that organizational optimization has become a lower priority during the COVID-19 pandemic. Third, the focus on Lean Thinking may have resulted in few outcome and structure measures found. A broader search strategy focusing on clinical pathways and types of measures used could have provided different insights but would potentially have resulted in too many measures outside the Lean Thinking scope of this review.

One of the strengths of this study is the connection made between process and outcome measures used in literature to analyze clinical pathways and the seven wastes, flow and pull of Lean Thinking. This has resulted in a practical framework that offers clear quantifications to analyze Lean Thinking in disease-oriented organizations. Our results show that optimization studies often only select one or two wastes or tools from Lean Thinking to analyze or implement in one part of the system, while our framework focuses on all the wastes analyzed in the entire system.

### 4.6. Implications

In this review, we have learned that Lean Thinking is not often quantified in scientific literature. With the quantifications suggested in this study, clinical pathways can be analyzed from a Lean Thinking perspective and in addition, a before and after measurement can be carried out for interventions. Further research is needed to study the use of the suggested framework for the analysis of an operational clinical pathway with real-time data. This study can investigate the practicality of the suggested quantifications and strengthen them with possible refinements. With a prospective cohort study, the current state of the clinical pathways can be measured with our suggested framework and the state after an improvement is implemented. The wastes, flow and pull can be compared to have a quantified result of the implementation.

Besides the implications for research, the framework can be beneficial for healthcare professionals. The current state of the clinical pathway can be used by healthcare professionals to search for bottlenecks and deviations from the desired pathway. Through this information, alterations can be made on how these bottlenecks or deviations can be prevented.

The remaining question is how more evidence-based management research can be performed. How can Lean Thinking in a healthcare setting be properly investigated?

## Figures and Tables

**Figure 1 healthcare-11-02488-f001:**
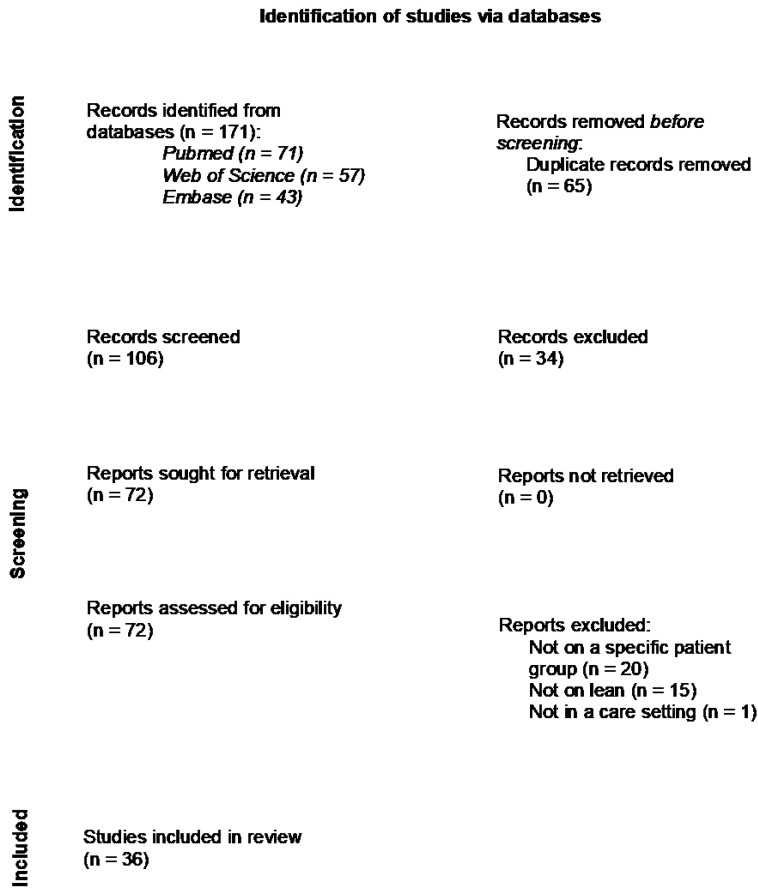
PRISMA flow diagram with review decision process [15].

**Figure 2 healthcare-11-02488-f002:**
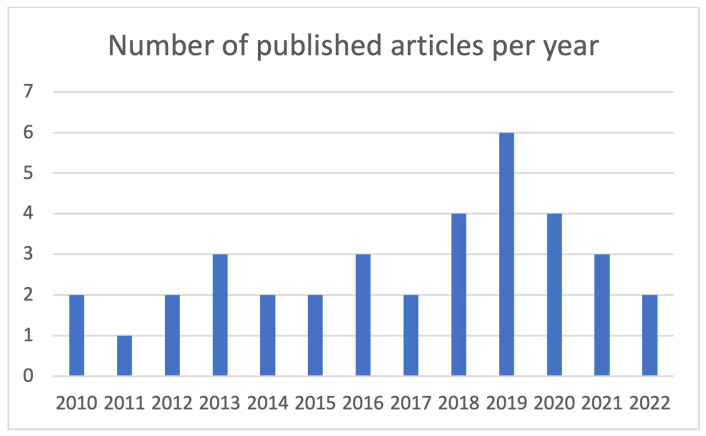
Total number of published articles per year.

**Figure 3 healthcare-11-02488-f003:**
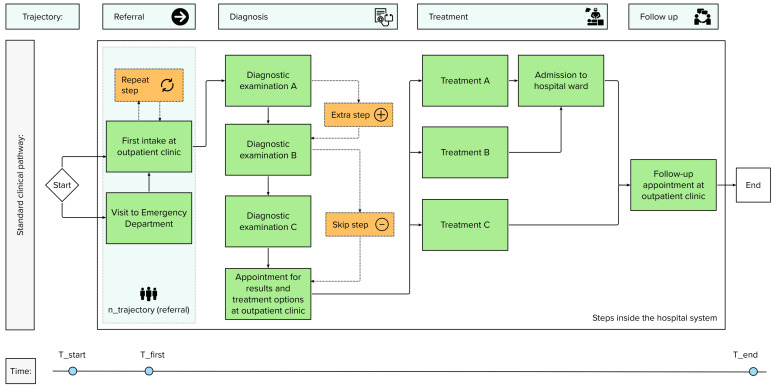
Example of a clinical pathway with different trajectories and steps.

**Figure 4 healthcare-11-02488-f004:**
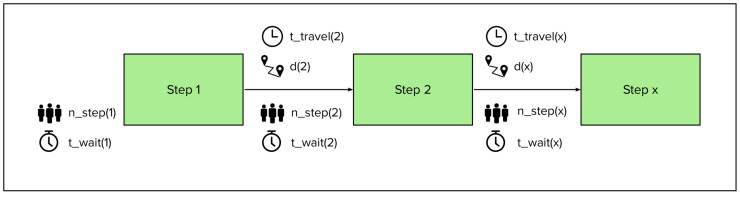
Zoom of Figure 3 with three steps as an example of variables measured per step.

**Figure 5 healthcare-11-02488-f005:**
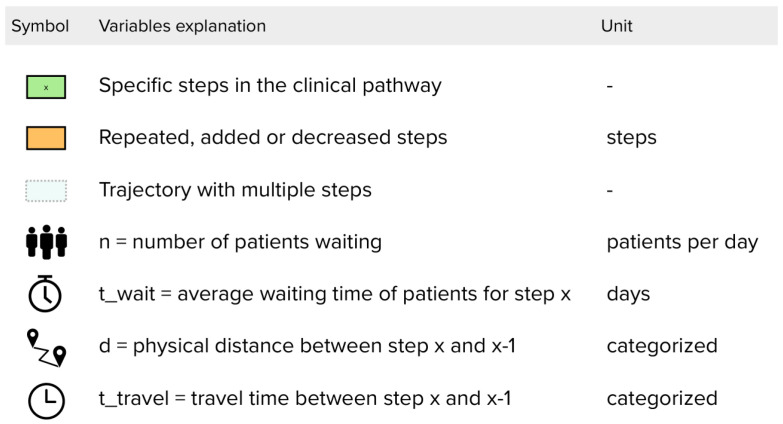
Legend of symbols used in Figure 3 and Figure 4.

**Table 1 healthcare-11-02488-t001:** Definitions of the seven wastes according to Liker [5].

Waste	Definition Liker
Overproduction	Producing items for which there are no orders, which generates such wastes as overstaffing and storage and transportation costs because of excess inventory.
Waiting	Workers merely serving to watch an automated machine or having to stand around waiting for the next processing step, tool, supply, part, etc., or just plain having no work because of stock-outs, lot processing delays, equipment downtime, and capacity bottlenecks.
Conveyance or unnecessary transport	Carrying work in process (WIP) long distances, creating inefficient transport, or moving materials, parts, or finished goods into or out of storage or between processes.
Overprocessing or incorrect processing	Taking unneeded steps to process the parts. Inefficiently processing due to poor tool and product design, causing unnecessary motion and producing defects. Waste is generated when providing higher-quality products than is necessary.
(Excess) inventory	Excess raw material, WIP, or finished goods causing longer lead times, obsolescence, damaged goods, transportation and storage costs, and delay. In addition, extra inventory hides problems such as production imbalances, late deliveries from suppliers, defects, equipment downtime, and long setup times.
Motion/unnecessary movement	Any wasted motion employees have to perform during the course of their work, such as looking for, reaching for, or stacking parts, tools, etc. In addition, walking is waste.
Correction or defects	Production of defective parts or correction. Repair or rework, scrap, replacement production, and inspection mean wasteful handling, time, and effort.

**Table 2 healthcare-11-02488-t002:** Medical (sub)specialties studied in the included articles.

Medical (Sub) Specialty ^#^	Number of Articles	Percentage of Articles
Internal medicine: Cardiology	3	8%
Internal medicine: Gastroenterology	1	3%
Internal medicine: Geriatric medicine	2	6%
Internal medicine: Infectious disease	1	3%
Internal medicine: Pulmonary disease	2	6%
Neurological surgery	1	3%
Neurology	2	6%
Ophthalmology	3	8%
Orthopedic surgery	14	39%
Otolaryngology	1	3%
Pediatrics	1	3%
Plastic surgery	2	6%
Psychiatry	1	3%
Surgery: General surgery	1	3%
Surgery: Transplant surgery	1	3%

^#^ List of specialties according to the American Board of Medical Specialties [16].

**Table 3 healthcare-11-02488-t003:** Lean methods used in the articles to optimize the process.

“4 P” Model	Lean Method	Number of Articles	Percentage of Articles ^#^
Philosophy	-	-	-
Process	Value stream mapping	19	53%
	Process mapping	12	33%
	Waste reduction	5	14%
	SIPOC	4	11%
	Standardization of Work	3	8%
	Continuous Flow	2	6%
	Single-piece flow	1	3%
People and Partners	-	-	-
Problem Solving	Cause-and-Effect Diagram	7	19%
	5 Whys	4	11%
	A3	2	6%
	Kaizen	2	6%
	Gemba Walk	1	3%
	Pareto	1	3%
	PICK Chart	1	3%

^#^ Total is more than 100% because multiple methods could be selected per article.

**Table 4 healthcare-11-02488-t004:** Types of process and outcome measures.

Type of Process Measures	Number of Articles	Percentage of Articles ^#^
Length of stay in hospital	16	44%
Time between process steps	13	36%
Duration of process steps	10	28%
Number of patients in certain process steps/trajectory	7	19%
Number of certain process steps in trajectory	6	17%
Pathway/guidelines compliance	3	8%
Duration of the whole trajectory	2	6%
Identification of bottlenecks (long tasks commonly performed)	2	6%
Treatment rate	2	6%
Protocol utilization	1	3%
Count of wastes in process	1	3%
Number of staff interactions	1	3%
Percentage of cancellations	1	3%
**Type of outcome measures**	**Number of articles**	**Percentage of articles ^#^**
Patient satisfaction	5	14%
Readmissions	5	14%
Complications	3	8%
Extra treatment needed	3	8%
Medication requirements	3	8%
Pain scores	3	8%
Mortality	2	6%
Perioperative scores	2	6%
Acute care utilization	1	3%
Discharge to which location	1	3%
Functional recovery	1	3%
**Type of cost measures**	**Number of articles**	**Percentage of articles ^#^**
Total cost	3	8%
Cost reductions	1	3%

^#^ Total is more than 100% because multiple measures could be collected in one article.

**Table 5 healthcare-11-02488-t005:** Link between found measures and Lean Thinking.

Lean Thinking Concept	Found Measures	Clarification
Overproduction	-	-
Waiting	Time between process steps	Waiting can be expressed as time a patient is waiting between process steps.
Transport	-	-
Overprocessing	Duration of process steps	On a process step level, if a process step takes longer than necessary, the step is overprocessed.
	Number of certain process steps in trajectory	On a process trajectory level, if the trajectory consists of too many steps, too little steps, or the wrong steps, then there is overprocessing.
	Pathway/guidelines compliance or protocol utilization	On a value stream map level, if the compliance/utilization is low, the wrong steps or too little/many steps are performed.
Inventory	Number of patients in certain process steps/trajectory	Inventory is defined, among other things, as work in process (WIP). The number of patients that are in a certain part of the value stream map, patients in process, can be seen as stock at a process trajectory level.
Movement	-	-
Defects	Length of stay in hospital	The duration of hospitalization is often seen in medical literature as an outcome measure. A hospitalization that is longer than usual is regarded as a defect (an undesirable outcome).
	Percentage of cancellations	The cancellation of an appointment or another process step is an undesirable outcome and often needs a correction (rescheduling).
	Patient satisfaction	If a patient is not satisfied with the given care, it has an undesired outcome.
	Readmissions	If a patient needs to be readmitted, it means that rework is taking place.
	Complications	A complication is not the correct outcome of care. It leads to extra work and is a defect.
	Extra treatment needed or extra medication required	If additional treatment is required, the previous treatment has not had the correct outcome and rework is required.
	Pain scores	An unnecessarily high pain score is an undesired outcome.
	Mortality	The mortality is an undesirable outcome and thus a defect.
	Acute care utilization	A visit to the emergency department could be prevented by providing care earlier. Acute care utilization is undesirable and a defect.
	Functional recovery	Functional recovery indicates whether the desired outcome is met with the care process. If the recovery is worse than desired, this is a defect.
Flow	Duration of the whole trajectory	When flow occurs, patients move smoothly through the value stream map. The duration of the whole trajectory can be an indication of flow, where a shorter duration would indicate more flow.
Pull	-	-
No link	Identification of bottlenecks (long tasks commonly performed)	A bottleneck is defined here as a process step that takes a long time and is performed frequently. Identifying these bottlenecks can be used to level the process (*heijunka*) and thus create flow. But a bottleneck is not a quantification of flow itself.
	Treatment rate	The percentage that a certain treatment took place in the hospital is not a measure that can be linked to Lean Thinking in a clinical pathway.
	Count of wastes in process	Because this measure has no further specification on how it is counted, it cannot be linked to quantifying Lean Thinking.
	Number of staff interactions	In Lean Thinking there is no valuing of more or less staff interactions.
	Discharge to which location	The location to which a patient is discharged can be seen as an outcome measure, but it is outside the scope of the process within the hospital.
	Total cost	Cost is often a measure in Lean Thinking to express the effect of improvements but is not seen as one of the wastes.
	Cost reductions	Cost is often a measure in Lean Thinking to express the effect of improvements but is not seen as one of the wastes.

**Table 6 healthcare-11-02488-t006:** Framework for measuring Lean Thinking in a clinical pathway. Suggested variable or formula linked to the Lean elements.

Lean Element	Description	Variable or Formula
Overproduction	Number of people waiting for step (x) per day.	n_step(x)
Waiting	Average waiting time of patients before step (x) in days for chosen period.	t_wait(x)
Conveyance	Physical distance to step x from step x-1, categorized as e.g., zero (for same location or digital), at department, in building, outside building.	d(x)
Overprocessing	Steps that deviate from the desired clinical pathway established by healthcare professionals.	s_repeat s_added s_skipped
Inventory	Total number of patients waiting in a certain trajectory per day.	n_trajectory(x)
Defects	E.g., patient satisfaction of the entire pathway, complications, mortality, and functional recovery.	defect_satisfaction defect_complication defect_mortality defect_PROM defect_PREM
Flow	Ratio between total waiting time and cycle time (time between first meeting to end of trajectory) per patient.	$\frac{t_{wait,1}+t_{wait,2}+t_{wait,x}}{t_{end}-t_{first}}$
Pull	Total number of patients waiting in clinical pathway per day.	$\sum_{i=1}^{n}n\_step(x_{i})$

## Data Availability

The data presented in this study are available in Appendix A.

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
