# Peer review of "How to Use Lean Thinking for the Optimization of Clinical Pathways: A Systematic Review and a Proposed Framework to Analyze Pathways on a System Level"

_healthcare, 2023, doi:10.3390/healthcare11182488_

Round 1

Reviewer 1 Report

Thank you for your submission.

The title of the manuscript is not really reflective of the actual content of the manuscript; where is the evidence presented that “Many hospitals embrace Lean Thinking only partially”?

Was the systematic review protocol registered with Prospero?

More details should be provided of the medical areas covered in the literature, rather than simply ”Patients from the medical specialty orthopedic surgery were most often 192 studied (n=14, 39%). The other specialties occurred one to three times.” Also, there should be some discussion around why there is a dominance in orthopedic surgery. It’s fascinating that orthopedic surgery is often likened to mechanical or manufacturing processes …perhaps that’s why the Lean principles have been transferred to that context? 

The definitions derived by the researchers in their proposed framework seem somewhat contrived/artificial and more related to convenience rather than a logical rationale related to Lean.

e.g. “Overproduction is producing items for which there are no orders, resulting in stock build up between processes and producing more than is required. To measure overproduction in a clinical pathway, we suggest using the number of patients waiting in the system between two steps. In the case of colorectal cancer, an example of overproduction is patients waiting for an operation.”    I can’t really understand this logic. Overproduction would be more like over-treatment (unnecessary treatment) of patients, in contradiction of best-practice guidelines, contributing to clogging of the healthcare system.

More problematic is “Pull: Total number of patients waiting in clinical pathway per day.”  This makes no sense in a Lean model. Pull refers to something like timely hospital discharges from hospital to create bed capacity for new admissions.

The English is mainly fine.

Reviewer 2 Report

The study presents valuable insights and contributes meaningfully to the existing body of knowledge in the field.

While the content of your research is relevant, there are areas, particularly in the presentation and certain methodological clarifications, that would benefit from revision. These refinements will not only enhance the readability and clarity of your work but will also align it more closely with the expectations and standards of a Q2 journal.

I would recommend revisiting the manuscript with these considerations in mind.

Looking forward to seeing the revised manuscript and hoping for a successful collaboration.

The overall quality of the English is commendable. However, I've highlighted a few sections that I believe need further review and refinement. Please go through the entire text to ensure grammatical accuracy and enhance the quality of the writing.
